# Endoglin Trafficking/Exosomal Targeting in Liver Cells Depends on *N*-Glycosylation

**DOI:** 10.3390/cells8090997

**Published:** 2019-08-28

**Authors:** Steffen Meurer, Almut Elisabeth Wimmer, Eddy van de Leur, Ralf Weiskirchen

**Affiliations:** RWTH University Hospital Aachen, Institute of Molecular Pathobiochemistry, Experimental Gene Therapy and Clinical Chemistry, D-52074 Aachen, Germany

**Keywords:** exosomes, lipid raft, Caveolin-1, endoglin, BMP, TGF-β, liver, hepatic stellate cells, hepatocytes, portal myofibroblasts, shedding, fibrosis

## Abstract

Injury of the liver involves a wound healing partial reaction governed by hepatic stellate cells and portal fibroblasts. Individual members of the transforming growth factor-β (TGF-β) superfamily including TGF-β itself and bone morphogenetic proteins (BMP) exert diverse and partially opposing effects on pro-fibrogenic responses. Signaling by these ligands is mediated through binding to membrane integral receptors type I/type II. Binding and the outcome of signaling is critically modulated by Endoglin (Eng), a type III co-receptor. In order to learn more about trafficking of Eng in liver cells, we investigated the membranal subdomain localization of full-length (FL)-Eng. We could show that FL-Eng is enriched in Caveolin-1-containing sucrose gradient fractions. Since lipid rafts contribute to the pool of exosomes, we could consequently demonstrate for the first time that exosomes isolated from cultured primary hepatic stellate cells and its derivatives contain Eng. Moreover, via adenoviral overexpression, we demonstrate that all liver cells have the capacity to direct Eng to exosomes, irrespectively whether they express endogenous Eng or not. Finally, we demonstrate that block of *N*-glycosylation does not interfere with dimerization of the receptor, but abrogates the secretion of soluble Eng (sol-Eng) and prevents exosomal targeting of FL-Eng.

## 1. Introduction

The different members of the transforming growth factor-β (TGF-β) superfamily, which include TGF-β, activins and bone morphogenetic proteins (BMPs), are regulatory cytokines involved in a multitude of biological processes [1]. In the liver, the prototype of this family, i.e., TGF-β1, is produced in parenchymal and non-parenchymal cells. During liver insults pivotal targets of TGF-β1 are hepatic stellate cells (HSC) and portal fibroblasts which become activated and acquire a myofibroblast-like phenotype, thereby inducing the profibrogenic program [2]. Due to the pleiotropic effects, the function of this cytokine is strictly regulated. TGF-β1 is stored inactively as a large latent complex in the extracellular matrix. It is proteolytically activated and released and its biological activity is modulated by many soluble factors [3,4]. Once released, the ligand docks to receptors located in the plasma membrane, a process which is tightly controlled by secreted proteins such as the connective tissue growth factor (CTGF) [5].

TGF-β signaling is initiated by binding of the ligand to a receptor complex, which contains type I (TβRI), type II (TβRII) cell surface receptors [6]. The TβRIs (ALK1 or ALK5) and TβRIIs are serine/threonine kinases with a similar structure containing a cysteine-rich extracellular domain, a short hydrophobic transmembrane domain, and a cytoplasmic region containing the kinase domain. Upon binding to TβRII, TβRI is recruited and becomes trans-phosphorylated by TβRII. Subsequently, the activation of the type I receptor results in endocytosis of the receptor complex, followed by phosphorylation of substrate proteins by TβRI. In general, there are two different ways of internalization involving clathrin coated pits (ccp, EEA-1 positive) or Caveolin coated pits (lipid rafts, Caveolin-1 positive). It is assumed that localization to ccp leads to Smad activation, whereas routing to lipid rafts leads to degradation of receptors or activation of non-Smad signaling [7,8]. Activated Smads form complexes with the common Smad4, which then translocate to the nucleus. In association with positive and negative transcription factors, Smads regulate a wide spectrum of target genes [9].

In addition to the signaling receptors TβRI and TβRII, there are two type III receptors, i.e., betaglycan and endoglin (Eng, also known as CD105), which modulate binding to the signaling receptors. Both receptors are single membrane spanning, glycosylated accessory receptors for members of the TGF-β superfamily and have no intrinsic kinase activity [10,11]. In case of Eng, defective *N*-glycosylation has been shown to interfere with membrane localization similar to TβRII [12,13]. For betaglycan it has been shown that changes in the type of glycosylation affect the signaling outcome [14]. Betaglycan is broadly expressed and has been shown to positively regulate ALK5/Smad2/3-signaling [15,16]. In contrast, Eng expression is more restricted, being highly expressed on endothelial cells and dampening ALK5/Smad2/3, while enhancing ALK1/Smad1/5 signaling [17]. Differential splicing of Eng creates a shorter variant and imposes an even higher degree of fine tuning to the signaling cascade [18]. Nevertheless, the exact mechanism(s) on how Eng governs this fine tuning is not known, but it is known to involve a direct interaction of Eng with TβRI and TβRII. We could previously demonstrate this interaction of Eng and TβRII in HSC [19]. In endothelial cells, Eng association with TβRI (ALK5 and ALK1) and TβRII, as well as the consecutive phosphorylation events on Eng, have been analyzed in detail. The corresponding consequences for response outcomes have been described and affect Smad and non-Smad signaling [20]. However, if Eng modulates those responses by affecting TβRI and TβRII routing is not known, since Eng trafficking modalities have been only sparsely analyzed so far [21,22].

The in vivo importance of Eng for signaling of individual ligands of the TGF-β family becomes evident by the human disease hereditary hemorrhagic telangiectasia type-1 (HHT-1) caused by Eng haploinsufficiency/and dominant negative effects caused by truncation, nonsense and missense mutants of Eng affecting endothelial cell biology [23,24]. On a mechanistical basis, mutations cause varying defects which abolish Eng function. Some missense mutations are not capable of ligand binding, while others are dominant negative by sequestering wild type Eng in a heteromeric complex inside the cell [12,25,26]. Several missense mutations cause misfolding, leading to premature proteins defective in final glycosylation most likely due to retention in the ER by the ERAD system [12,26].

In addition to the membrane bound receptor, soluble Eng (sol-Eng) has been implicated in HHT-1 [27]. The extracellular domain of Eng (sol-Eng) can be released from the cell by a regulated shedding mechanism involving matrix metalloprotease-14 (MMP-14) [28]. The shedded receptor has been initially shown to contribute to the pathogenesis of pre-eclampsia and the sol-Eng content in the serum of pre-eclamptic women is increased [29]. Although it is claimed that Eng only binds ligands in the presence of signaling receptors, sol-Eng is secreted, can dimerize, is able to directly bind to BMP9 and BMP10, and affects TGF-β1 signaling [29,30].

Recently, it was demonstrated that exosomes isolated from the serum of pre-eclampsia patients contain abundant sol-Eng, contributing to the damage of vascular functions and complications in pre-eclampsia patients [31]. Exosomes are nano-sized membrane vesicles in size from 30–150 nm formed by the inward budding of late endosomes that are released into the extracellular environment. They can contain a diverse array of signaling molecules (e.g., proteins, mRNAs, and microRNAs) and are key players in mediating cell paracrine effects by transferring genetic materials and proteins to target cells [32]. In regard to the liver, exosomes represent one of the communication routes between the parenchymal cells (PC, hepatocytes) and mesenchymal cells (HSC, pMF) and vice versa [33]. HSC-derived exosomes contain components, which induce activation of quiescent HSC and metabolic activity of Kupffer cells (KC) and liver sinusoidal endothelial cells (LSEC) to promote fibrosis [34].

In addition to LSEC in the liver, HSC produce comparable amounts of Eng and its expression during hepatic fibrogenesis is upregulated—especially in HSC [19,20,35]. Genetic depletion of Eng in HSC leads to an aggravation of fibrosis, implying Eng as an antifibrotic receptor [36]. However, to our best knowledge, there are no reports, showing that full-length Endoglin (FL-Eng) is cargo of exosomes in general and especially in HSC.

We show here that Eng is enriched in Caveolin-1 containing fractions (lipid rafts). In addition, in this study we isolated exosomes from different cellular primary sources of the liver and a number of different immortalized hepatic cell lines. We could show that exosomes derived from primary rat HSC or immortalized Col-GFP cells derived from primary mouse HSC contain large quantities of FL-Eng. Likewise, the human cell line LX-2, originally established by spontaneous immortalization in low serum conditions, had the potential to direct Eng to exosomes when transiently infected with an adenoviral vector expressing FL-Eng or sol-Eng. Moreover, exosome localization did not depend on endogenous Eng expression, since hepatocyte-derived exosomes were positive for heterologously expressed Eng—although PC lack endogenous Eng. Finally, we show that block of *N*-glycosylation does not inhibit sol-Eng or FL-Eng dimerization but does impede sol-Eng secretion and FL-Eng exosomal localization.

## 2. Materials and Methods

### 2.1. Cloning and Purification of Adenoviral Expression Vectors

Cloning of recombinant adenoviruses for the expression of FL-Eng and sol-Eng was done in the AdEasy-1 system [37]. In brief, the respective cDNA fragments of FL- (2230 bp) and sol-Eng (1789 bp) included in vectors pcDNA3.1-FL-Endoglin [35] and pcDNA3.1-sol-Endoglin [38] were first cloned into the shuttle vector pShuttle-CMV vector [37]. Therefore, the vector pcDNA3.1-FL-Endoglin was digested with restriction sites *Kpn*I (Roche, Mannheim, Germany) and *Pme*I (New England Biolabs GmbH, Frankfurt am Main, Germany) and the resulting fragment cloned into vector pShuttle-CMV cut with *Kpn*I and *EcoR*V (Roche). The fragment of pcDNA3.1-sol-Endoglin was isolated by cutting with *Kpn*I and *Not*I (Roche) and cloned into the shuttle vector digested with the same enzymes. Resulting plasmids were purified and verified by sequencing. For homologous recombination, the resultant plasmids were linearized by digestion with restriction endonuclease *Pme*I, and co-transformed with the adenoviral backbone vector pAdEasy-1 into BJ5183 bacteria. Recombinants were selected for kanamycin resistance, and recombination was confirmed by restriction analysis and sequencing. For the generation of recombinant adenoviruses, the recombinants were produced by transfection into the mammalian embryonic kidney packaging cell line 293A [39] using Lipofectamine (Life Technology, ThermoFisher Scientific, Darmstadt, Germany) as the transfection agent and following the protocol described elsewhere [37]. Viruses were harvested and for higher titer viral stocks, packaging cells were repeatedly infected. The final purification of adenoviruses was performed on a cesium chloride density gradient using ultracentrifugation following by CsCl banding following further purification using the Virapur Adeno-X™ Virus Purification Kit (#K1654-2), which was formerly obtained from Clontech (Palo Alto, CA, USA) and is now distributed by Takara Bio Europe (Saint-Germain-en-Laye, France).

### 2.2. Culturing of Immortalized Cell Lines

Human HSC cell line LX-2 [40], human hepatoma cell line HepG2 [41], the immortalized mouse HSC reporter line Col-GFP [42], the rat HSC line CFSC [43], the Chinese hamster cell line CHO-Lec 3.2.8.1 [44], and African green monkey COS-7 (ATCC, Manassas, VA, USA, CRL-1651) cells were cultured in Dulbecco’s modified Eagle’s medium (DMEM) with high glucose (#D6171) supplemented with 10% fetal calf serum (FCS, #F7524), 5 mM sodium pyruvate solution (#S8636), 2 mM L-Glutamine solution (#G7513), 1 x Penicillin/Streptomycin solution (#P0781) all obtained from Sigma-Aldrich (Taufkirchen, Germany). Cell passaging was done using Accutase cell detachment solution (#A6964) from Sigma-Aldrich. In addition, CFSC medium was supplemented with nonessential amino acids (Life Technology, ThermoFisher Scientific).

### 2.3. Isolation and Culturing of Primary Cells from Rat and Mouse

Portal fibroblasts from rats were isolated from male Sprague-Dawley rats using standardized protocols published before [45]. In brief, rats weighing 300–500 g were anesthetized with ketamine and xylazine followed by exploratory laparotomy. The liver was perfused in situ with collagenase solution. After removal of the hepatic hilum, the remaining tissue parts were digested serially with a 0.25% pronase solution (Merck, Darmstadt, Germany) and 0.1% hyaluronidase (Sigma-Aldrich). After filtering through a 100 µm-pore mesh, cells were cultured for one day in DMEM/F12 medium (#21331-020, Gibco, ThermoFisher Scientific) supplemented with 3% FCS and 2 mM L-Glutamine, and 1 x Penicillin/Streptomycin solution. Thereafter, the serum content was increased to 10% FCS for prolonged culturing. Cells at passage number from 3 to 5 were taken for experimentation.

Primary HSC from rat and mouse were routinely prepared by collagenase/pronase digestion of liver using a perfusion system and subsequent fractionation of the heterogenous cell suspension on continuous density gradients made out of Nycodenz. Experimental details on the isolation procedure are given elsewhere [46,47].

### 2.4. Remarks on Ethical Issues on Animal Experimentation

Permission for the isolation of primary cells from livers of rats and mice was granted by the Landesamt für Natur- und Umweltschutz (LANUV, Recklinghausen, Germany). Permission date of respective approval with identification no. 84-02.04.2015.A028 was given on 4 March 2015.

### 2.5. Stimulation of Cells

Cells were plated in six-well dishes or 10 cm plates and left untreated or were adenovirally infected (see Section 2.8). Before stimulation, the cells were starved for 16 h in medium containing 0.5% FCS. Thereafter, the medium was changed to medium containing 0.5% FCS, and after 1 h ligand (rhTGF-β1, Cat. No. 240-B-002, R&D Systems, Wiesbaden, Germany) was applied for the indicated times and concentrations. Cells were washed three times in 1x PBS, scraped off the well in RIPA lysis buffer containing 20 mM Tris-HCl (pH 7.2), 150 mM NaCl, 2% (*w*/*v*) NP-40, 0.1% (*w*/*v*) SDS, 0.5% (*w*/*v*) sodium deoxycholate including protease and phosphatase inhibitors and prepared for Western blot as described under Section 2.13.

### 2.6. Transient Transfection

Cells were plated in six-well dishes at a density of 2–3 × 10^5^ cells per well. The next day, the medium was exchanged and 1 h later the cells were transiently transfected using Lipofectamine2000 according to the manufacturer’s instructions. Briefly, DNA/Lipofectamine2000 complexes were set up in OptiMEM using a ratio of 2:4 (µg DNA/µL Lipofectamine 2000), added to the cell medium and incubated for 24 h. Thereafter culture medium was exchanged and HepG2 or HSC Col-GFP cells were either left for a further 24 h and the supernatant was used for co-immunoprecipitation (HepG2, Section 2.12) or after 10 h medium was exchanged and cells were starved for 16 h and stimulated the next day (HepG2, HSC Col-GFP, Section 2.5). For COS-7 cells the medium was exchanged after transient transfection and cells were cultured for further 24 h before the addition of tunicamycin for 24 h, followed by Western Blot analysis (Section 2.13)

### 2.7. Sucrose Gradient Analysis

Sucrose density gradient analysis was performed at 4 °C. Cells (CFSC or primary rat HSC) were grown to near confluence in 10 cm dishes. After washing with ice-cold phosphate-buffered saline (PBS), the cells were scraped into 0.85 mL of sodium carbonate buffer (500 mM, pH 11.0). The cell pellets were homogenized in an ultrasonic disintegrator (model UP100H from Hielscher Ultrasound Technology, Teltow, Germany) with three 20-s bursts on ice. The homogenates were adjusted to 45% sucrose using 0.85 mL 90% sucrose in MES-buffered saline (25 mM MES, 150 mM NaCl, pH 6.5) and placed at the bottom of an ultracentrifuge tube (Beckman Coulter). Two solutions (1.7 mL each) of 35% and 5% sucrose were laid sequentially on the top of the 45% sucrose solution. After ultracentrifugation at 39,000 rpm in a Beckman Optima^TM^ L-70K centrifuge using a SW 40 Ti rotor for 20 h, 11 individual 0.5-mL fractions were collected from the top of the tubes, and 50 µL of each fraction were analyzed by Western blot (Section 2.13).

### 2.8. Adenoviral Infection

Twenty four hours after seeding, the serum content of the medium was reduced from 10% to 4%. The cells were then infected with a MOI of 2 for 12 to 24 h. Thereafter, the content of serum was re-adjusted to 10% and cells were grown for indicated time points. As a control, we used an adenovirus construct directing expression of luciferase under transcriptional control of the CMV promoter.

### 2.9. Isolation and Characterization of Exosomes from Conditioned Media

Exosomes from conditioned media was essentially done following an ultracentrifugation protocol that was published before [48] and modified by our group [45]. Briefly, the conditioned media of untreated, adenoviral or mock-infected cells were harvested and centrifuged at 600× *g* at 4 °C in a BS 4402/A rotor for 15 min in a Heraeus Sepatech refrigerated centrifuge. The resulting supernatant was collected and centrifuged at 3200× *g* at 4 °C for 30 min to remove remaining cell fragments. The second supernatant was then filtered through a 0.22 μm sterile syringe filter (#431219, Corning GmbH, Kaiserslautern, Germany) and the cleared solution was then centrifuged at 100,000× *g* (29,500 rpm; RCF_avg_ 109,895; RCF_max_ 154,779; k-factor: 2525) at 4 °C for 70 min in a Beckman Optima^TM^ L-70K ultracentrifuge equipped with a SW 40 Ti rotor. The supernatant was carefully removed, and the crude exosome-containing pellets were washed once in ice-cold 200 mM 4-(2-hydroxyethyl)-1-piperazineethanesulfonic acid (HEPES) buffer (pH 7.0) and pooled. A second round of ultracentrifugation under the same conditions was carried out, and the final pellet highly enriched in exosome particles resuspended in 200 μL of 200 mM HEPES buffer (pH 7.0) for Western blot analysis (Section 2.13).

### 2.10. Tunicamycin Treatment

One day after initial plating, the culture medium of transfected (Section 2.6) or infected (Section 2.8) cells were replaced (or not, if sol-Eng was analysed) with fresh medium containing 0.5 μg/mL tunicamycin from *Streptomyces* sp. (#T7765, Sigma-Aldrich). Cells were then cultured for an additional 24 h period. Thereafter, the conditioned cell culture media were harvested and cell protein extracts prepared for Western blot analysis. Cells cultured for the same time without tunicamycin or in the presence of vehicle (dimethyl sulfoxide) were taken as controls.

### 2.11. Precipitation of Glycosylated Proteins with ConA Beads

For the precipitation of glycosylated proteins, we used beads coupled to the lectin Concanavalin A (ConA, *Canavalia ensiformis*, Cat. No. 234568, Calbiochem). Before performing the binding reaction, the beads were washed three times with culture medium including 10% FCS if proteins from the supernatant were used for the reaction or RIPA buffer if proteins from the lysates were precipitated each for 15 min at 4 °C using a rotating overhead. Thereafter, 250 µL (from 2 mL of one six-well) of the corresponding cell culture supernatants or 100 µg of the lysates of COS-7 or HSC Col-GFP cells were added to the beads in a total volume of 500 µL and rotated overnight at 4 °C. The next day, complexes were washed three times with culture medium, including 0.5% FCS three times, each 15 min at 4 °C rotating overhead, resolved in 20 µL RIPA buffer /20 µL 4x LDS sample buffer (ThermoFisher Scientific, #NP0007) and 2 µL (1 M) dithiothreitol (DTT) and analysed by Western blot (Section 2.13).

### 2.12. Co-Immunoprecipitation of Endoglin and TGF-β1 with PPabE2 from Supernatants of sol-Eng and Mock-Transfected HepG2

For co-immunoprecipitation, 500 µL cell culture supernatant of mock- (pcDNA) and sol-Eng-transfected HepG2 cells were supplemented with 50 ng TGF-β1 and mixed at 4 °C for 2 h using a rotating overhead. Thereafter, 2 µg of rat Eng-specific antibody PPabE2 [19] were added and the samples were further rotated for 2 h followed by addition of Protein G Plus-Agarose (Protein G Plus-Agarose, Cat. no. sc-2002, Santa Cruz Biotech., Santa Cruz, CA, USA) and rotation overnight at 4 °C. The next day, the complexes were washed twice with 300 µl of culture medium each 15 min rotating at 4 °C. Finally, complexes were resolved in 20 µL RIPA/20 µL 4x LDS sample buffer, supplemented with DTT and used for Western blot analysis (Section 2.13).

### 2.13. Sodium Dodecylsulfate Polyacrylamide Gel Electrophoresis and Western Blot Analysis

Cell lysates were prepared in RIPA buffer and containing the Complete™ mixture of proteinase inhibitors (Roche) and Phosphatase inhibitor cocktail 2 (#P5726, Sigma-Aldrich). Equal amounts of total cellular proteins (10–50 µg/lane) or exosome proteins (~2 µg/lane) determined by the DC protein assay (Bio-Rad, Düsseldorf, Germany) were mixed with NuPAGE™LDS electrophoresis sample buffer (Invitrogen, ThermoFisher Scientific) supplemented with DTT as a reducing agent or left untreated. Denaturation of protein samples, electroblotting and Western blot analysis were essentially applied as described previously [42]. After blotting, the transfer of proteins to the membrane was shown in Ponceau S stain and unspecific binding sites were blocked with 5% (*w*/*v*) non-fat milk powder in Tris-buffered saline with Tween 20 (TBST). For detection of individual proteins, the primary antibodies were diluted in 2.5% (*w*/*v*) non-fat milk powder in TBST. All antibodies used in this study are listed in Table 1 and their specificity for proteins of mouse, rat and human origin tested (Appendix A). Primary antibodies were visualized with anti-mouse, anti-rabbit or anti-goat IgG secondary antibodies (all from Santa Cruz Biotech, Santa Cruz, CA, USA) with the SuperSignal chemiluminescent substrate (Pierce, Bonn, Germany).

## 3. Results

### 3.1. FL-Endoglin Is Co-Localized with Caveolin-1 in the Lipid Raft Fraction

TGF-β superfamily signaling is initiated by binding of the ligand to a surface exposed receptor complex. Upon binding, the receptors become activated and are endocytosed to perform the signal transfer to effector substrates, e.g., Smads or non-Smad substrates. As described above, there are two different routes for internalization of TGF-β receptors, the ccp and lipid rafts enriched in Caveolin-1. Since the outcome of the TGF-β-response as well as the further routing to exosomes is critically influenced by the subdomain in which the receptor is situated in, we first analyzed the localization of Eng in membrane subdomains of HSC. Therefore, we separated cell protein extracts either prepared from HSC cell line CFSC or primary rat HSC by sucrose gradient centrifugation (Figure 1). In the CFSC separation, Caveolin-1 was presented in fraction 5–8 (Figure 1A). For Eng, the separation was more complex with the appearance of the mature glycosylated protein and the dimeric form only in lipid rafts, whereas a low molecular weight species, most likely less glycosylated protein, was found in the non-raft membrane fractions (Figure 1A). In the HSC separation, Caveolin-1 was distributed in the fractions 4–8 (Figure 1B). Eng was present in the mature glycosylated and the dimeric form. However, in contrast to the finding obtained with CFSC, it was distributed over the lipid raft and non-lipid raft fractions with a higher abundance in the caveolin-1 enriched fractions. These results demonstrate that Eng is localized to Caveolae, but not exclusively since the receptor is also present in non-lipid raft fractions (Figure 1A,B). Since we were focusing on Eng exosomal trafficking, we also had a look at the exosomal marker presence in the membrane subdomains. Interestingly, in both cell types, the lipid rafts were nearly devoid of Alix, while CD81 was exclusively localized to caveolae (Figure 1A,B).

### 3.2. Cloning of an Adenoviral Vector for Expression of Rat Full-Length Endoglin

In order to have a proof-of-principle, we first sought to overexpress FL-Eng to analyse its exosome localization to avoid sensitivity problems due to its mutual sparse exosomal abundance. As a control, we also included sol-Eng in our initial experiments, because sol-Eng exosomal localization is already established [27]. Based on the fact that primary HSC and portal myofibroblast (pMF) are nearly inaccessible to standard transfection procedures [49], we constructed adenoviral constructs using the Ad-Easy-1 system for expression of FL-Eng and sol-Eng. These vectors are directed under the control of the ubiquitous CMV minimal promoter/enhancer driving effective expression of the 650 amino acid long FL-Eng (Figure 2) and the shortened 576 amino acid encoding soluble form lacking the transmembrane domain (TM) and the cytosolic domain (CD) comprising the whole extracellular domain in direct proximity to the established cleavage site (Figure 3, Appendix A). As a prerequisite, we evaluated the specificity of antibodies for the detection of exosomal marker proteins, TGF-β-receptors, corresponding substrates, and target proteins in human HSC cell line LX-2, rat primary HSC, and mouse HSC reporter cell line Col-GFP (Appendix A).

In order to check the functionality of Eng upon viral overexpression, we performed stimulation experiments and analysed selected target proteins [19,36,42]. In primary cells (rat pMF and rat HSC) and permanent cells (LX-2), FL-Eng is highly expressed upon infection. In pMF and LX-2, Eng enhances the TGF-β1-mediated α-smooth muscle actin (α-SMA) and CTGF expression. In rat HSC, those markers were already expressed at a very high level so that Eng overexpression did not provoke a visual difference of signals (cf. Figure 2C–E). These results confirm the effects we published for mouse HSC Col-GFP and rat CFSC-2G using transient transfection for FL-Eng overexpression and further implies that the viral background has no impact on the experimental outcome [19,42].

### 3.3. Cloning of an Adenoviral Vector for Expression of Rat Soluble Endoglin

So far, we only analysed FL-Eng function in liver cells. Therefore, we sought to validate the functionality of the soluble construct before using it for exosome localization studies. For the soluble construct, we chose the complete extracellular domain missing two amino acids in direct proximity to the determined cleavage site which leads to shedding of Eng from the plasma membrane (Figure 3A,B). This extracelluar domain has been shown to be secreted, produce dimers and binds ligand [20]. In a first test COS-7 cells, as well as CHO cells, were transiently transfected with empty vector or sol-Eng (Figure 3C). In both cell types, sol-Eng could not be detected in the lysates (most likely due to a co-migrating, cross-reactive protein, see below), but yields a prominent band in the supernatants. In general the expression efficiency for sol-Eng in CHO was higher compared to COS-7 cells. Omitting DTT in the samples resulted in a sol-Eng monomer and a dimer band, indicating that the expressed protein is secreted and capable of forming dimers. Next, we transiently transfected the hepatocyte derivative HepG2 with a sol-Eng cDNA (Figure 3D). Those parenchymal cells lack endogenous Eng and would therefore be potential paracrine targets of mesenchymal cell-derived sol-Eng. sol-Eng was detected in lysates of HepG2 cells and its presence reduces the expression of the TGF-β1 target protein CTGF at all-time points analysed. In order to test if sol-Eng can bind ligand, we next incubated supernatants from mock-and sol-Eng-transfected HepG2 with recombinant TGF-β1 and precipitated sol-Eng thereafter. In the precipitated complexes, we found in addition to sol-Eng also TGF-β1 implying that TGF-β1 binds to sol-Eng (Figure 3E). Finally, we transiently transfected the HSC line Col-GFP with empty vector or sol-Eng (Figure 3F). Similar to the results obtained in COS-7 and CHO cells, sol-Eng could not be detected in lysates but was secreted in the supernatants. Both concentrations of TGF-β1 that we used (0.1, 1.0 ng/mL) in this set of experiments caused an up-regulation of the fibrogenic marker proteins α-SMA and CTGF. sol-Eng increased the expression of those markers at both concentrations of TGF-β1 (Figure 3F). A similar effect of sol-Eng on secreted CTGF in the supernatant is seen at higher ligand concentrations.

In summary, the experiments show that the virus-mediated FL-Eng transfer is equivalent to the transient transfection delivery (Figure 2) and that the sol-Eng construct is functional active and can be used to direct viral overexpression (Figure 3).

### 3.4. Identification of Exosomal Marker Proteins

In our laboratory, we have recently established a method for isolating and characterizing exosomes from supernatants of cultured cells using a protocol that is based on ultracentrifugation [45]. Previous studies have demonstrated that the extent of exosome release and composition is cell-dependent [50]. Since we used several immortalized cell lines and primary cells in our study, we first tested for the occurrence of typical exosomal markers (CD81; Alg2-interacting protein, Alix; Caveolin-1) within the different cell populations used for experimentation (Figure 4, Table 2).

This analysis showed that the repertoire of exosomal makers purified from cells of hepatic origin can be highly variable. Moreover, the only ubiquitous marker found was Alix, while CD81 and Cav-1 were expressed in mesenchymal cells, but not in HepG2 cells. Cav-1 expression in HSC is very low in rat HSC and undetectable in our experiments in mouse HSC. Nevertheless, we cannot exclude that the expression is below our detection level.

### 3.5. Full-Length and Soluble Endoglin Are Targeted to the Exosome Compartment of Permanent Cell Lines

To get a first impression if Eng is targeted to exosomes, we isolated exosomes from the human HSC cell line LX-2 and the hepatocyte derived hepatoma cell line HepG2. Therefore, both cell lines were infected with the respective adenoviral vectors for FL- and sol-Eng or a control vector directing expression of luciferase (Luc) under control of the CMV promoter. Exosomes of LX-2 cells show the typical exosomal markers Alix, CD81 and Cav-1 (Figure 5A). In contrast, exosomes derived from HepG2 cells (Figure 5B) contain only Alix, but not CD81 and Cav-1. With respect to Eng, virus-mediated expression in the lysates of LX-2 and HepG2 is very prominent. This is especially an improvement for sol-Eng when compared to the transient transfection in COS-7, CHO or HSC Col-GFP cells (cf. Figure 5A,B and Figure 3C,F). In addition to the lysates, there is a high abundance of sol-Eng in the supernatants pre- and post-centrifugation, confirming an efficient secretion of the extracellular domain of Eng. Nevertheless, although MMP-14 is present in LX-2 and HepG2 lysates (not shown), we failed to detect the shedded sol-Eng in FL-Eng-infected cells. Most interestingly, FL-Eng and to a lesser extend sol-Eng were found in the exosomal fraction (Figure 5A,B). The differential expression of exosomal markers and the presence of Eng was even more evident when exosomal fractions were separately analysed in Western blot (Figure 5C). In the next step, we examined the cell line HSC Col-GFP, which has been shown to express endogenously Eng [42]. In the corresponding exosomes there is a high enrichment of CD81 and a moderate enrichment of Alix, but only very sparse amounts of Cav-1 (Figure 5D). Endogenous FL-Eng is found in exosomes of these HSC derivatives. Stimulation with TGF-β1 for 48 h induces FL-Eng expression in general in lysates and in line increases its inclusion in exosomes. In addition to FL-Eng, we could not detect TβRI (ALK5) or TβRII and only very low amounts of TβRIII in exosomes of HSC Col-GFP (Figure 5A,B,D).

### 3.6. Full-Length Endoglin Is Targeted to the Exosome Compartment of Primary Mesenchymal Liver Cells

In the process of liver fibrosis mesenchymal cells, including HSC and pMF play a key role in providing pro-fibrogenic mediators to surrounding cells and shift the ECM homeostasis to matrix accumulation by increasing the synthesis and decreasing the degradation [51]. We have shown earlier that FL-Eng is highly expressed in primary HSC and pMF of rat origin [35]. Therefore, we sought to determine if FL-Eng is also localized in exosomes of primary cells. When analyzing exosomes derived from rat HSC, we found a prominent expression of CD81 in this fraction and only a marginal expression of Alix (Figure 6A,B). In contrast, Cav-1 was not detectable in exosomes of rat HSC, although expressed at a high level in those cells as demonstrated by its presence in the corresponding lysates. In order to prove exosomal inclusion in those primary cells, we traced endogenous expressed as well as virally overexpressed FL-Eng (Figure 6A) This set of experiments revealed that both the lower expressed endogenous FL-Eng and the overexpressed one were detectable in exosomes. However, although MMP-14 is expressed in HSC (Figure 5C), we were not able to detect sol-Eng in supernatants or exosomes, although the transient overexpression forced a dramatic increase in FL-Eng expression. Finally, we analysed endogenous FL-Eng trafficking to exosomes in primary rat pMF (Figure 6C). In line with primary HSC there is a strong enrichment of CD81 and Cav-1—in contrast to HSC—in exosomes. Similar to HSC, Alix is expressed at a very low level. The presence of endogenous FL-Eng could clearly be demonstrated in all three experiments. Nevertheless, in primary rat pMF ALK5 (TβRI) and β-glycan (TβRIII) are detectable in the exosomal fraction.

### 3.7. Glycosylation of Endoglin

Eng is a transmembrane glycoprotein with four verified *N*-linked glycosylation sites and a potential O-linked glycosylation site rich in serines and threonines [11,12,20]. We next wanted to test whether *N*-linked sugar modifications in Eng are functionally important for trafficking and secretion of the artificially-constructed sol-Eng and exosomal targeting of FL-Eng. Several missense mutations in FL-Eng of patients suffering from HHT-1 get stuck in the endoplasmic reticulum due to misfolding, which most likely in turn leads to an aberrant glycosylation pattern [12]. In order to test whether missing *N*-glycosylation affects trafficking of Eng, we decided to concentrate on *N*-glycosylation, which can be blocked in cells by pretreatment with tunicamycin, a strong inhibitor of *N*-linked glycosylation and which has been used for FL-Eng deglycosylation before [12]. We have previously shown that this inhibitor blocked glycosylation of the soluble protein Lipocalin 2 (LCN2), but does not hamper secretion or exosome targeting [45]. As a functional consequence, tunicamycin induces the unfolded protein response in cells, which we could demonstrate for COS-7 cells by expression of the marker protein Chop (Appendix A), as well as Bip and Chop for HSC Col-GFP (Appendix A). To validate the effect of tunicamycin on Eng, we first transfected COS-7 cells with an expression vector for FL-Eng and as a further control TβRIII (betaglycan), which is also known to be highly *N*-glycosylated [15]. For the tunicamycin treatment, we chose a low concentration of the inhibitor (0.5 µg/mL) and a time period of 24 h, which should result in the appearance of both glycosylated and lower glycosylated species, due to the published FL-Eng half-life of ~17 h [23]. The treatment of the transfected cells with tunicamycin resulted in a significant loss of highly glycosylated TβRIII and a molecular shift of core band that was detectable in Western blot analysis (Figure 7A, left panel). In contrast, tunicamycin causes a loss of glycosylated FL-Eng nearly without the appearance of a lower molecular weight species (Figure 7A, right panel, Appendix A). The difficulty in detecting the deglycosylated FL-Eng and sol-Eng monomers in lysates (which migrate at a similar molecular weight) arises from a cross-reactive protein migrating at the same molecular weight (Appendix A). The transfection of the respective plasmid into the Chinese hamster cell line CHO-Lec 3.2.8.1 carrying mutations that give rise to severely truncated *N*- and O-linked carbohydrates [44] resulted in a significant molecular shift of FL-Eng, most likely representing the unglycosylated protein. However, also in this cell line, tunicamycin causes a loss of FL-Eng. In sum, tunicamycin blocked glycosylation of TβRIII with a loss of heterogeneously highly glycosylated receptors and a shift of the core band to a lower molecular weight, whereas glycosylated FL-Eng expression was reduced without the appearance of a lower molecular species. These results were confirmed by ConA precipitation of the glycosylated receptors (Figure 7B). Both receptors are glycosylated and are therefore bound by ConA, whereas after tunicamycin treatment, there is a loss of TβRIII binding as well as part of the low molecular weight core stays unbound due to defective glycosylation (Figure 7B, left). In contrast FL-Eng binding was lost, but there was no deglycosylated receptor present in the unbound fraction (Figure 7B, right panel).

On a more functional level, we examined the secretion of sol-Eng under these conditions. Therefore, supernatants were analysed for the heterologously expressed sol-Eng in the presence or absence of DTT (Figure 7C). As shown before (Figure 3C), sol-Eng presented as a dimer and monomer in the absence and as monomer in the presence of DTT. Moreover, there was no lower migrating band detectable, which would indicate a deglycosyled form. The lower expression was confirmed by ConA precipitation (Figure 7D). In contrast, when analyzing HSC Col-GFP there was a clear molecular weight shift of FL-Eng seen in the lysates upon tunicamycin treatment (Figure 7E). Mature and deglycosylated proteins appeared in nearly equimolar amounts. As shown before for sol-Eng, mature FL-Eng is assembled as a DTT-sensitive dimer, which is also true for the deglycosylated FL-Eng.

According to the literature [52], TβRII was also shifted in molecular weight, indicating that tunicamycin treatment prevented *N*-glycosylation of this receptor. In line, the activation of Smad2 was reduced in the presence of tunicamycin (Figure 7E) confirming a previous report [13]. ConA precipitation of proteins isolated from HSC reporter cell line Col-GFP that were treated or not with tunicamycin resulted in binding of mature but not deglycosylated FL-Eng. Due to the higher amount of deglycosylated FL-Eng in HSC Col-GFP, this protein is found in the unbound fraction (Figure 7F). However, in case of TβRI and TβRII, both mature and lower glycosylated forms were found in the bound and to a much lower degree in the unbound fractions (Figure 7F). Finally, we evaluated the secretion of sol-Eng in Col-GFP via heterologous overexpression. sol-Eng appeared as a dimer and monomer in the absence as well as monomer in the presence of DTT, and a lower expression level upon tunicamycin treatment, but without any indication of a lower molecular weight species. In sum, these experiments imply that (i) FL-Eng and sol-Eng are able to form dimers upon removal of *N*-glycosylation but (ii) sol-Eng secretion is abrogated.

### 3.8. Proper Glycosylation of Endoglin Is Necessary for Exosome Targeting

We next addressed the question if proper glycosylation is necessary for exosome targeting as it is essential for secretion of the sol-Eng (Figure 7C,D,G). Therefore, we infected a first set of experiments LX-2 cells with FL-Eng or as a control with Luc and treated cells with TGF-β1 or tunicamycin for 24 h. Whereas we only detected a very faint amount of the lower glycosylated FL-Eng and sol-Eng in COS-7 cells (Appendix A), we found a prominent molecular weight shift of FL-Eng upon treatment with tunicamycin in the lysates of LX-2 cells (Figure 8A). In line, tunicamycin leads to a molecular shift of TβRII and TβRI. But except FL-Eng, none of the other examined TGF-β receptors were found in the exosomes of LX-2. However, analysis of the exosomal compartment indicates that only the glycosylated form of Eng is targeted to the exosome. A similar result was obtained when primary mouse HSC were analysed (Figure 8B). Beside FL-Eng, ALK5 also shows a molecular shift indicative for the loss of glycosylation. Nevertheless, only fully glycosylated FL-Eng was targeted to exosomes in line with the LX-2 results.

## 4. Discussion

TGF-β1 is a key player in the pathogenesis of liver disease involving the activation of HSC and pMF as well as the induction of their fibrogenic program, leading to a wound healing reaction. Aside the TGF-β signaling receptors TβRI and TβRII, which are both indispensable for signaling, FL-Eng representing an accessory receptor for TGF-β1 is highly expressed in HSC and upregulated during activation of HSC [35]. In addition, the overexpression FL-Eng in HSC cell lines causes a higher expression of the activation marker α-SMA and the profibrogenic protein CTGF [19,42]. In contrast, genetically manipulated mice lacking FL-Eng in HSC show a higher fibrogenic response in the setting of liver intoxication (carbon tetrachloride) and cholestasis (bile duct ligation). However, siRNA-based or adenovirally-expressed Cre-mediated knockdown of FL-Eng in primary isolated HSC results in reduced CTGF expression [36].

To solve these discrepant findings, we tried to understand the mechanism of Eng function in more detail. Although membrane localization of Eng has been addressed in a plentitude of studies concerning with hereditary hemorrhagic telangiectasia type 1 (HHT-1), there is only very limited information available about Eng trafficking in liver cells. Since the localization of receptors to membrane subdomains (i.e., caveolae or ccp) is vital for signaling outcome, we first sought to locate Eng in those subdomains via sucrose density gradient analysis. In murine endothelial cells, this analysis revealed a coupling of Eng and eNOS in caveolae to regulate eNOS stability and activity. In these studies, Eng was exclusively localized to lipid rafts [53]. In contrast, our analysis showed a distribution of Eng in lipid rafts as well in non-raft fractions. However, there is a higher abundance of Eng in raft fractions compared to non-raft fractions (Figure 1). The importance of Caveolin-1 (and therefore lipid rafts) for HSC activation and biology has been emphasized in recent study by others, showing that overexpression of Caveolin-1 in a HSC cell line (GRX) affected divers functions of HSC including activation, cell cycle, profibrogenic behavior and migration [54]. Therefore, we are currently investigating whether Eng overexpression or depletion in HSC as well as depletion of cholesterol changes the localization and distribution of the signaling receptor TβRI and TβRII and finally TGF-β-signaling in those subdomains.

Extracellular vesicles are lipid-bilayer enclosed vesicles secreted by virtually all cell types contributing to cellular communication by shuttling of a large variety of cargo molecules [55]. They play key roles in both normal and disease processes and can influence biological functions in cells located in close vicinity or at distant locations via transfer through the systemic circulation [55].

In our study, we prepared “exosomes” which have a typical size of 30–100 nm and were sedimented at 100,000× *g* [45]. However, it should be critically mentioned that the usage of the term “exosome” is somewhat vague and should be used with caution [56]. Based on the suggestions published by the International Society for Extracellular Vesicles (ISEV) the term exosome can be used in three different ways [57]. Some researchers argue that the term exosome refers to vesicles that bud into endosomes and are released when the resulting multivesicular bodies fuse with the plasma membrane. Others argue that exosomes are secreted vesicles that may have a physiological function, while the third researchers classify exosomes as particles that only sediment after centrifugation at ~70.000–100.000× *g*.

In regard to liver, several biological functions have been attributed to exosomes. Hepatocyte-derived exosomes from primary human hepatocytes were recently shown to promote liver immune tolerance [58]. Another report has recently shown that cholangiocyte-derived exosomes enriched by the noncoding RNA-H19 enhance transdifferentiation of cultured mouse primary HSC and promote progression of cholestatic liver fibrosis [59]. Similarly, the delivery of exosomes released from activated HSC can provoke metabolic switches in nonparenchymal liver cells affecting glucose metabolism by delivery of glycolysis-related proteins [34]. Exosomes derived from HCC cell lines include many proteins, microRNAs, long noncoding RNAs, mRNAs, and DNAs [50]. Therefore, it was proposed that some exosomes may be potential diagnostic biomarkers for early-stage hepatocellular carcinoma (HCC) [60].

On the other hand, there is growing evidence that exosomes function as conduits for the intercellular transfer of components to induce resolution of hepatic fibrosis by inhibiting macrophage activation, cytokine secretion, modulation of extracellular matrix, and inactivation of HSC [33]. Experimentally, it was shown that delivery of miRNA targeting CTGF can suppress fibrogenic signaling in human HSC [61]. It is obvious that exosomes are therefore potentially of fundamental importance for the therapy of fibrotic liver lesions and to interfere with processes relevant in the pathogenesis of HCC.

This is underpinned by the finding that microvesicles derived from Eng-positive cancer stem cells can confer an activated angiogenic phenotype to normal human endothelial cells and stimulate proliferation and vessel formation [62], which is also a key process in HCC.

In our study, we tested whether individual liver cells form exosomes containing endogenously expressed Eng or have the capacity to direct overexpressed Eng to the exosomal compartment. Therefore, we have cloned adenoviral expression constructs expressing either FL-Eng or sol-Eng (Figure 2 and Figure 3, Appendix A) and tested for the functionality of FL-Eng and sol-Eng in comparison with our previously published results and literature data. In addition, we tested if proper glycosylation of Eng is a mandatory need to be secreted for the artificial sol-Eng and targeted to exosomes for FL-Eng. In respective experiments, we either prevented proper glycosylation in liver cells by treatment with tunicamycin which inhibits *N*-linked glycosylation by preventing core oligosaccharide addition to nascent polypeptides, or alternatively by overexpressing FL-Eng in the Chinese hamster cell line CHO-Lec 3.2.8.1, allowing to produce glycoproteins in which all *N*-linked carbohydrates in the Man5 oligomannosyl form and O-linked carbohydrates are truncated to a single GalNAc [44].

We first isolated and characterized exosomes in regard to the existence of typical exosomal marker proteins from several primary cultured liver cells and immortalized derivatives thereof. Interestingly, the subset of exosomal markers (CD81, Alix, Cav-1) usually recognized as exosomal markers were not found in all liver cells. In contrast to a previous report [63], CD81, which belongs to the group of endosome-specific tetraspanins having a broad tissue distribution and known to be enriched in exosomal membranes [64] was not detected in HepG2 cells in our study (Figure 3, Table 2). However, other cells tested (LX-2, Col-GFP, rat HSC, mouse HSC, rat pMF) expressed large amount of this marker. Alix, which is involved in exosome biogenesis and exosome sorting and common to most exosomes, was found in all liver cells tested (Figure 4, Figure 5 and Figure 6 and Figure 8). Cav-1, a multifunctional membrane protein typically upregulated in the final stages of cancer and associated with promotion of migration and invasion [65] was found to be expressed in LX-2, Col-GFP and rat pMF, while in primary HSC from rat and mouse, as well as human HepG2, this scaffolding protein was virtually absent.

Primary rat HSC, Col-GFP representing a cell line derived from murine HSC, and rat pMF showed high endogenous expression of Eng, confirming our previous reports [19,20,35]. Nevertheless, viral-mediated overexpression leads to an even higher abundance of FL-Eng in these cells (Figure 2C–E). Similarly, sol-Eng is highly expressed in LX-2 and HepG2 upon viral transduction. Respective cells were also capable to target FL-Eng and sol-Eng to exosomes (Figure 5A and Figure 6E).

In contrast, HepG2 cells, which lack endogenous Eng, were able to direct adenovirally expressed FL-Eng and sol-Eng to exosomes (Figure 5). This suggests that the endosomal sorting complex required for transport (ESCRT) during exosome biogenesis [57] of all liver cells have the capacity to sort Eng into exsomes. However, although we could show exosomal localization of sol-Eng by overexpression of the truncated FL-Eng, we were not able to detect shedded sol-Eng originating from the expressed FL-Eng (Figure 5). This is unexpected, since the effector protease is expressed in the analysed cells and is co-localized with FL-Eng in exosomal fraction (Figure 4 and Figure 5) [36,66].

In previous work from other groups, defined glycosylation signatures identified by glycomic analysis were shown to be enriched in exosomes [67]. Based on the fact, that Eng contains four verified *N*-linked glycosylation sites and a potential O-linked glycosylation site [11,12], we next tested if glycosylation of Eng is mandatory for exosome targeting. We found that the blockade of *N*-glycosylation by tunicamycin resulted in a significant molecular shift of FL-Eng in gel electrophoresis indicative of lower glycosylation in primary mouse HSC and LX-2 (Figure 8). Due to the experimental setting, both fully and low glycosylated species are detected. However, only fully glycosylated FL-Eng is found in the exosomal fraction, but not the lower molecular weight species (Figure 8). Lux et al. showed that all four *N*-glycosylation sites are used, and that eliminating one or two sites does not affect membrane targeting, but loss of three sites nearly abolishes surface expression [12]. Although exosomal localization of FL-Eng is lost when not properly glycosylated, the dimerization of this receptor still occurs (Figure 7E). Nevertheless this previous report has not given details about surface localization of tunicamycin treated FL-Eng. Based on our experiments, we cannot exclude that the unglycosylated FL-Eng does not reach the plasma membrane. However, a general arrest of proteins in the ER induced by tunicamycin can be excluded as at least LCN2 is secreted in both glycosylated and none glycosylated forms [45].

In line with the missing localization in exosomes of deglycosylated FL-Eng we were not able to demonstrate secretion of deglycosylated sol-Eng—neither in COS-7, nor in HSC Col-GFP (Figure 7C,G). This points strictly towards an intracellular sequestration of the deglycosylated Eng and a loss of function. For the signaling receptor TβRII, loss of proper *N*-glycosylation is associated with abolished receptor function [13], which we assume also for Eng. Nevertheless, the tunicamycin-induced unfolded protein response does not necessarily block TGF-β1 signaling as shown for HSC Col-GFP (Figure 7E, Appendix A). Moreover, it can also activate those responses in a cell type-specific manner [68]. In the case of COS-7 cells, those immature proteins potentially have a low half-life since it was difficult to display these lower migrating species. In contrast to COS-7 cells, the glycosylated and deglycosylated Eng forms are present in equal amounts after tunicamycin treatment, which implies that those immature receptors are stabilized in HSC Col-GFP.

In our view, the finding that Eng is part of extravesicular vesicles is extremely important. These particles have gained interest due to their roles in cell-to-cell communication and potential roles of exosomes are discussed for many liver diseases, including alcoholic fatty liver disease/non-alcoholic fatty liver disease [69], hepatitis infection [70], hepatocellular carcinoma [60], and liver metastasis [71]. Several aspects in these malignancies such as immune cell infiltration and fibrogenesis are majorly driven by members of the TGF-β superfamily. In addition, it is well accepted that exosomes isolated from women with preeclampsia can effectively transfer sol-Eng to endothelial cells, where it provokes significant cell damage [31]. In regard to liver, such mechanisms were not described before, but the eminent importance of Eng and exosomes in the setting of hepatic diseases is undoubtedly. Even more, there are no reports available, which systemically analyze different liver cells for their potential to secrete Eng-containing exosomes.

However, our study has some limitation because we have no data pointing to the function of exosomal Eng. The function of cargo-loaded Eng will of course depend on its absolute concentration. Other factors are the potential target cells, the physiological or pathological state of the organism or organ, concentrations of Eng ligands being present, the capacity to degrade them, and many other factors. We are aware of the fact that these questions must be answered in future studies.

So in summary, our study showed that: (i) Eng is present in lipid rafts; (ii) all liver cells have the capacity to direct Eng to exosomes, irrespectively if they express endogenous Eng; (iii) proper Eng *N*-glycosylation is mandatory for trafficking of Eng leading to secretion of sol-Eng and exosome targeting.

## 5. Conclusions

Fundamental studies during the last decades have shown that TGF-β signaling is a key function in the pathogenesis of hepatic fibrosis. A plethora of studies has unraveled molecular details by which TGF-β contributes to the initiation or progression of hepatic disease. This knowledge has offered potential targets for pharmacological intervention. The intricate network of TGF-β signaling is further increased by the identification of Eng endocytosis via the lipid rafts. Moreover, Eng is present in exosomes of liver cells. These particles are known to contribute to cell-to-cell communication by transferring active components from the releasing cell into a susceptible recipient cell. It is evident that exosomes derived from HSC might contribute to hepatic angiogenesis, which is closely associated with the progression of fibrosis in chronic liver disease and the formation of hepatocellular carcinoma.

## Figures and Tables

**Figure 1 cells-08-00997-f001:**
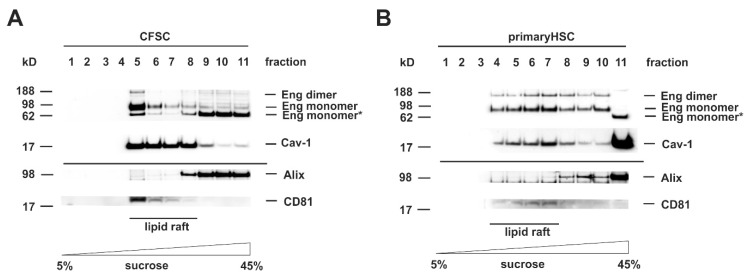
Endoglin is localized in Caveolin-1 enriched lipid rafts. (**A**) CFSC and (**B**) primary rat hepatic stellate cells (HSC) were cultured to near confluence under basal conditions (10% FCS). Thereafter, the cells were harvested in carbonate buffer, sonicated, and the extracts separated on a sucrose gradient. Eleven consecutive fractions were collected starting from the top (low sucrose) to the bottom (high sucrose). Proteins of the individual fractions (equal volumes) were separated by SDS-PAGE and analyzed by Western blot analysis. Lipid rafts were identified by detection of Caveolin-1. The presence of Endoglin (Eng), Alix, and CD81 were analyzed by specific antibodies.

**Figure 2 cells-08-00997-f002:**
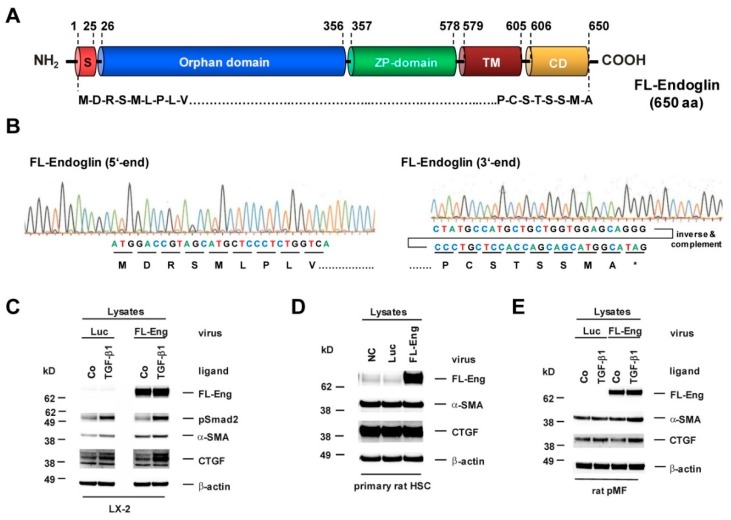
Structure, sequence analysis and functional expression of a virus coding for full-length Endoglin from rat. (**A**) The full-length endoglin (FL-Eng) form is composed of a signal peptide, an orphan domain, a zona pellucida (ZP) domain, a transmembrane region (TM), and a cytosolic domain (CD). Amino acid positions spanning individual regions are numbered. (**B**) Sequence analysis of FL-Eng expression plasmid. For clarity the sequence data of the 3′end was inversed and complemented for depicting the encoded amino acids. (**C**–**E**) Human LX-2, primary rat HSC and pMF were infected with a control virus (Luc), FL-Eng or left untreated (NC). Thereafter, cells were treated or not with TGF-β1 (1 ng/mL) for 24 h (**C**,**D**), 48 h (**E**) or cultured without any treatment (Co) (**D**). Cells were harvested and proteins analyzed using Western blot for transgene expression (FL-Eng), connective tissue growth factor (CTGF) and α-smooth muscle actin (α-SMA) or activation of Smad2 (pSmad2). Equal loading of lanes was proven by re-hybridization of the membranes with an antibody specific for β-actin.

**Figure 3 cells-08-00997-f003:**
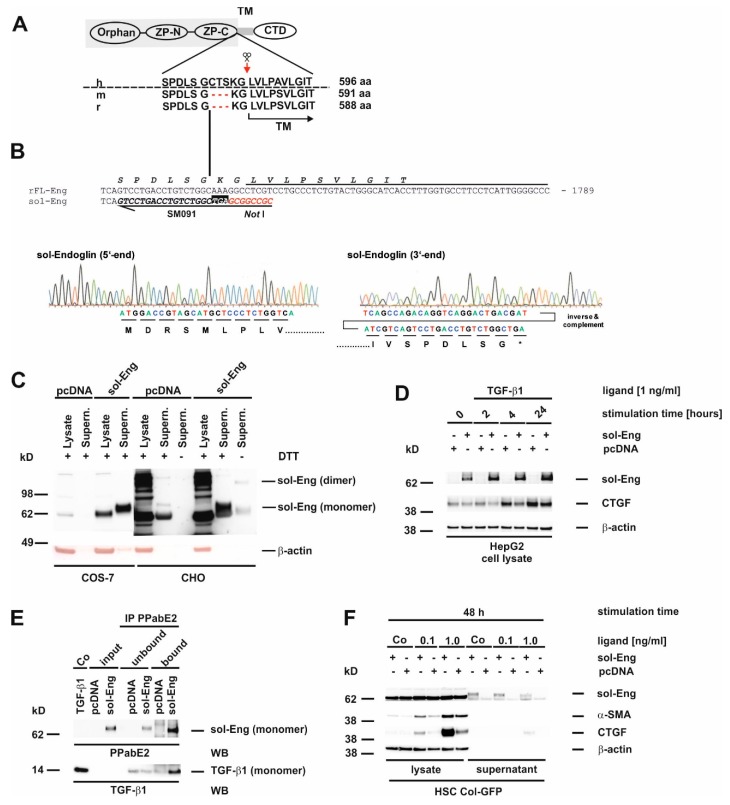
Structure, sequence analysis and functional expression of soluble Endoglin from rat. (**A**) Scheme of the FL-Endoglin (FL-Eng) domain structure including the amino acid environment surrounding the cleavage site (scissors and red arrow) of human, mouse and rat. (**B**) Nucleotide sequence of FL-Eng and the corresponding primer composition (SM091) to amplify sol-Eng and introduce a stop codon (TGA) and a *Not*I restriction site (upper panel). Sequence analysis of the sol-Eng expression plasmid (lower panel). For clarity the sequence data of the 3′end was inversed and complemented for depicting the encoded amino acids. (**C**) COS-7 cells (left) and CHO cells (right) were transiently transfected with empty vector (pcDNA) or sol-Eng expression vector (sol-Eng). sol-Eng expression was displayed in Western blot in the presence (+) or absence (−) of DTT. (**D**–**E**) HepG2 cells were transiently transfected using empty vector (pcDNA) or sol-Eng expression vector (sol-Eng). Thereafter, cells were either treated or not with TGF-β1 (1 ng/mL) for the indicated time points and sol-Eng expression and CTGF expression was monitored in Western blot (**D**) or the supernatant was supplemented with TGF-β1. Binding was accomplished for 2 h rotating and sol-Eng complexes were precipitated overnight using a rat Eng-specific antibody (PPabE2) and analysed in Western blot for the presence of bound TGF-β1 and precipitated sol-Eng. (**F**) HSC Col-GFP cells were transiently transfected using empty vector (pcDNA) or sol-Eng expression vector (sol-Eng). After starvation for 16 h, the cells were stimulated or not for 48 h with TGF-β1 (0.1, 1.0 ng/mL). Cellular proteins were analysed in Western blot for transgene expression (sol-Eng), CTGF and α-SMA. Equal loading of proteins was proven by re-hybridization of the membranes with a β-actin antibody.

**Figure 4 cells-08-00997-f004:**
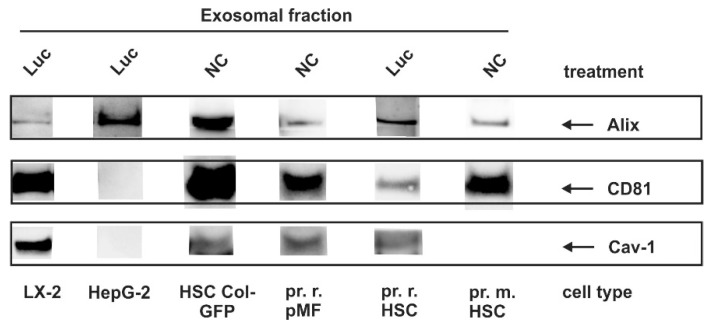
Exosomal marker expression in liver cells. Cells were plated on 10 cm dishes and either cultured in growth medium without further treatment (NC), or subjected to viral infection with a control virus (Luc). Thereafter, exosomes were prepared from the supernatants and marker expression was analyzed by Western blot using antibodies directed against Alg2-interacting protein (Alix), CD81 and Caveolin-1 (Cav-1).

**Figure 5 cells-08-00997-f005:**
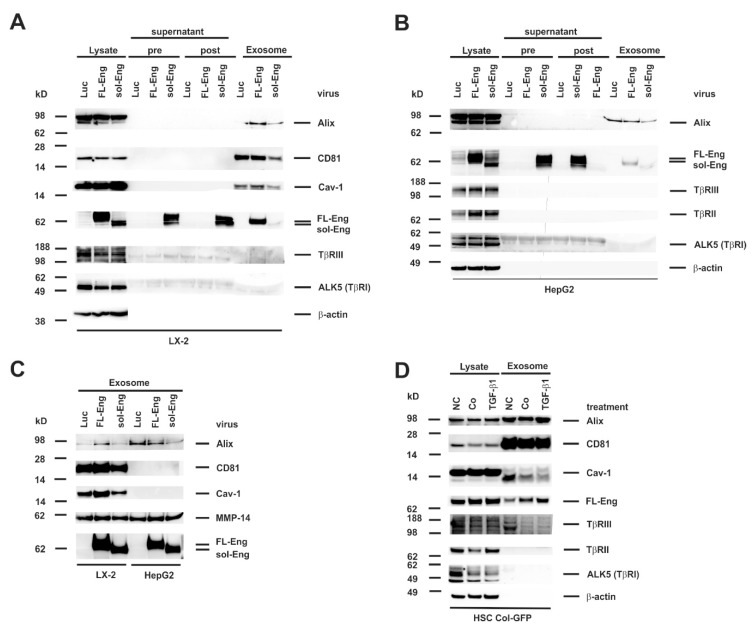
Identification of full-length Endoglin and soluble Endoglin in liver cell lines. (**A**,**B**) Cell protein extracts, supernatant (pre), wash solution (post), and final exosome preparations from (**A**) LX-2 or (**B**) HepG2 infected with adenoviral constructs expressing luciferase (Luc), full-length endoglin (FL-Eng), or soluble endoglin (sol-Eng) were analysed by Western blot for expression of indicated proteins. (**C**) Direct comparison of proteins included in exosomes purified from cultured LX-2 and HepG2 cells which were infected with adenoviral expression vectors driving expression of Luc, FL-Eng, and sol-Eng. (**D**) Cell lysates and corresponding exosomes fractions from HSC Col-GFP were analyzed in parallel in Western blot analysis. Exosomal markers and TGF-β-receptors were detected with specific antibodies. To monitor equal protein loading, the membrane was re-probed with a β-actin specific antibody. Please note, that only Endoglin and no other TGF-β-receptors analyzed were detected in significant amounts in exosomes. Abbreviations used are: NC, normal control meaning untreated cells cultured in the presence of normal growth medium containing 10% FCS; Co/TGF-β1, cells that were starved overnight (~16 h) in medium containing 0.5% FCS before stimulation with TGF-β1 (1.0 ng/mL) (TGF-β1) or not (Co) for 24 h.

**Figure 6 cells-08-00997-f006:**
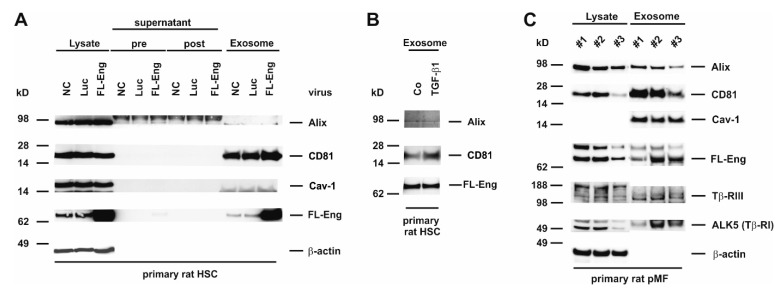
Identification of full-length Endoglin in primary mesenchymal liver cells. (**A**) Cell protein extracts, supernatant (pre), wash solution (post), and final exosome preparations from primary rat HSC untreated (NC), infected with adenoviral constructs expressing luciferase (Luc) or full-length endoglin (FL-Eng) were analysed by Western blot for expression of indicated proteins. (**B**) Primary rat HSC were stimulated or not with TGF-β1 (1 ng/mL) for 24 h and exosomes were prepared. The corresponding proteins were analysed by Western blot for the expression of exosomal marker proteins and FL-Eng. (**C**) Cell lysates and corresponding exosomal proteins of three independent experiments of primary rat pMF were analysed by Western blot for exosomal markers and TGF-β-receptors. For displaying equal protein loading, the membrane was reprobed with a β-actin specific antibody. Note that in contrast to other liver cell lines and primary cells, rat pMF express significant amounts of Tβ-RIII and Tβ-RI in exosomes.

**Figure 7 cells-08-00997-f007:**
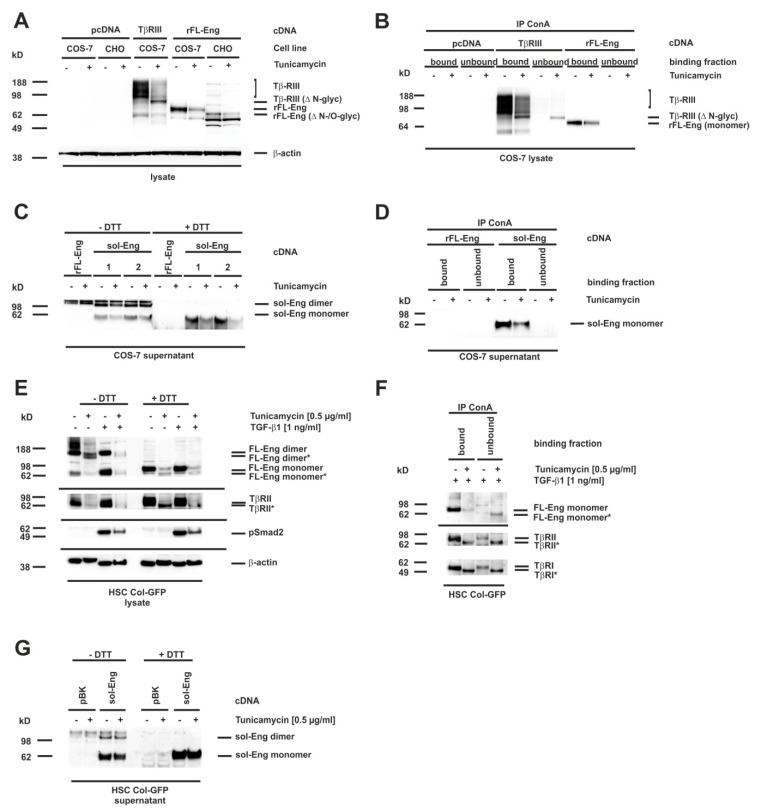
Blocking of *N*-glycosylation by tunicamycin leads to aberrant trafficking of Endoglin. (**A**–**D**) CHO or COS-7 cells were transfected with plasmids directing expression of rat FL-Endoglin (rFL-Eng), betaglycan (TβRIII), and rat soluble Endoglin (sol-Eng). A transfection with empty vector (pcDNA) served as control. Thereafter, cells were treated with tunicamycin (0.5 µg/mL, 24 h) or left untreated. (**A**) Cell extracts were prepared and analysed for expression of rFL-Eng and betaglycan. To monitor equal protein loading the membrane was reprobed with a β-actin specific antibody. (**B**) Cell lysates of COS-7 cells taken from (**A**) were incubated with ConA beads for precipation of glycosylated proteins. The ConA-bound fraction as well as the unbound fraction of the precipitates were then analysed by Western blot for rFL-Eng and betaglycan. (**C**) Supernatants of COS-7 cells transfected with vectors expressing rFL-Eng or sol-Eng were treated or not with tunicamycin (0.5 µg/mL, 24 h). Protein extracts were analysed by Western blot in the presence or absence of DTT (50 mM) for expression of sol-Eng. (**D**) Supernatants of COS-7 cells taken from (**C**) were incubated with ConA beads for precipitation of glycosylated proteins. The ConA-bound fraction as well as the unbound fraction of the precipitates were then analysed by Western blot for sol-Eng. (**E**) HSC Col-GFP cells were treated or not with tunicamycin (0.5 µg/mL, 24 h) and stimulated or not with TGF-β1 (1 ng/mL; 30 min). Thereafter, proteins were extracted and analysed by Western blot in the presence or absence of DTT (50 mM) for expression or activation of indicated proteins. Equal protein loading was demonstrated with a β-actin specific antibody. (**F**) Protein lysates of HSC Col-GFP (only TGF-β1 stimulated samples) taken from (**E**) were incubated with ConA beads for precipitation of glycosylated proteins. The ConA-bound fraction as well as the unbound fraction of the precipitates were then analysed by Western blot for the indicated receptor proteins. (**G**) HSC Col-GFP cells were transfected with a plasmid directing expression of rat soluble-Endoglin (sol-Eng) or empty vector control (pBK). Thereafter, cells were treated with tunicamycin (0.5 µg/mL, 24 h) or left untreated. Cell protein extracts were prepared and analysed in Western blot in the presence or absence of DTT (50 mM) for expression of sol-Eng.

**Figure 8 cells-08-00997-f008:**
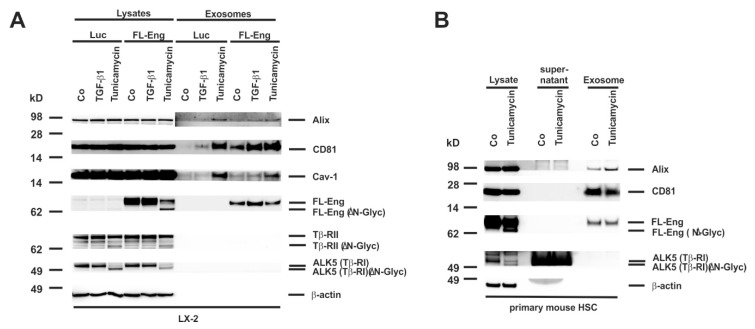
Isolation of exosomes from HSC treated with tunicamycin. (**A**) Cell extracts or exosome fractions prepared from LX-2 which were infected with FL-Eng or a control virus expressing luciferase (Luc). Subsequently, the cells were treated with TGF-β1 or an inhibitor of *N*-glycosylation (tunicamycin) and analysed by Western blot for expression of indicated proteins. Please note, that only properly glycosylated Eng is directed to the exosomal compartment. (**B**) Cell protein extracts, supernatants, and exosomes were prepared from primary mouse HSC and analysed for the presence of indicated proteins demonstrating that endogenous Eng is only directed to exosomes when properly glycosylated.

**Table 1 cells-08-00997-t001:** Antibodies used in this study in alphabetical order.

Antibody	Cat. No.	Clonality	Supplier/Reference	Species *	Dilution
Primary antibodies					
α-smooth muscle actin (α-SMA) (ASM-1)	CBL-171	mono (m)	Millipore, Merck, Darmstadt, Germany	h, m, r, b, ch, eq	1:2000
β-actin	A5441	mono (m)	Sigma, Taufkirchen, Germany	h, m, r, gp, can, hm, f, p, car, c, rb, sh, b	1:10,000
Alix (1A12)	sc-43540	mono (m)	Santa Cruz, Santa Cruz, CA, USA	h, m, r	1:1000
ALK5 (V-22)	sc-398	poly (rb)	Santa Cruz	h, m, r	1:500
Caveolin-1	CS-3238	poly (rb)	Cell Signaling Technology, Frankfurt am Main, Germany	h, m, r, ha, z, b, p	1:1000
CD81 (B-11)	sc-166029	mono (m)	Santa Cruz	h, m, r	1:1000
CTGF	sc-14939	poly (g)	Santa Cruz	h, m, r	1:1000
Endoglin	AF1320	poly (g)	R&D Systems, Wiesbaden, Germany	m	1:2000
MMP-14	AB38971	poly (rb)	Abcam, Cambridge, UK	h, m, p	1:1000
PPabE2	NA	poly (rb)	[19]	r	1:2000
pSmad2	CS-3101	poly (rb)	Cell Signaling Technology	h, m, r	1:1000
TGF-β1	CS-3711	poly (rb)	Cell Signaling Technology	h, m, r	1:500
TGF-βRII (L-21)	sc-400	poly (rb)	Santa Cruz	h, m, r	1:500
TGF-βRIII	AF242PB	poly (g)	R&D Systems	h	1:1000
Secondary antibodies					
IgG-HRP	sc-2004	NA	Santa Cruz	rb	1:5000
IgG-HRP	sc-2005	NA	Santa Cruz	m	1:5000
IgG-HRP	sc-2056	NA	Santa Cruz	g	1:5000

* Abbreviations used are: α-SMA—α-smooth muscle actin; a—avian; b—bovine; c—chicken; can—canine; car—carp; CTGF—connective tissue growth factor; eq—equine; f—feline; g—goat; gp—guinea pig; h—human; ha—hamster; hm—*Hirudo medicinalis*, m—mouse; mi—mink; mk—monkey; mono—monoclonal antibody; p—pig; poly—polyclonal antibody; NA—not applicable; o—ovine; p—pig; rb—rabbit; sh—sheep; z—zebrafish.

**Table 2 cells-08-00997-t002:** Exosomal markers identified in liver cells.

Cell Type						
	Immortalized Cell Lines			Primary Cells		
Surface Marker	HepG2	LX-2	Col-GFP	Rat HSC	Mouse HSC	Rat pMF **
CD81	−/− *	+	+	+	+	+
Alix	+	+	+	+	+	+
Caveolin-1	−/−	+	+	+	-	+

* Symbols represent: +, expressed in exosomes; −, not expressed in exosomes; −/− not expressed at all. ** Abbreviations used are: pMF, portal myofibroblast.

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
