# Peer review of "Endoglin Trafficking/Exosomal Targeting in Liver Cells Depends on N-Glycosylation"

_cells, 2019, doi:10.3390/cells8090997_

Round 1
Reviewer 1 Report
The authors have dealt adequately with my previous concerns.
Reviewer 2 Report
The authors addressed my concerns satisfactorily.
This manuscript is a resubmission of an earlier submission. The following is a list of the peer review reports and author responses from that submission.
Round 1
Reviewer 1 Report
This paper describes the expression and characterization of full length endoglin (FL-Eng) and a truncated mutant that behaves as soluble Endoglin (Sol Eng) in a number of (hepatic) cell lines after transduction with recombinant adenoviral vectors. Furthermore, a population of small extracellular vesicles, denoted exosomes, are isolated from these transduced cell lines or from primary hepatic stellate cells and myofibroblast and are shown to contain FL-Eng. Although the work looks sound, it is merely descriptive and offers only a very limited novel insights into the biology of these class III TGF-beta receptors.
Major points
1. The claim that the fraction isolated from the cells that contain the FL-Eng are really exosomes is not substantiated. The isolation method used, merely results in a class of small extravesicular vesicles of which exosomes may be a part. See ref Biogenesis, Secretion, and Intercellular Interactions of Exosomes and Other Extracellular Vesicles. By Marina Colombo, et al. Annu. Rev. Cell Dev. Biol. 2014. 30:255–289. The cited ref 41 also does not clarify this issue as it does not describe an exosome/EV isolation.
2 . The presence of a protein , in this case Eng, in an extravesicular vesicle without any additional data on the specificity/mechanism of routing to these vesicles and any functional relevance of the presence of this protein in the vesicles offers only a very limited advance in our knowledge, as EV are known to contain numerous proteins.
3. Page 8.Fig 2F. The authors claim in line 299-300 that “Sol-Eng reduced the expression of those markers [induced by TGF-beta i.e. alpha SMA and CTGF]” . In my opinion the blot 2F shows the opposite i.e a clear increase in alpha SMA and CTGF in the lanes were Sol-Eng is expressed.
Minor
Legend Figure 4 D on page 11 is incomplete. What are NC and Co and what concentration and incubation time of TGF-beta1?
Reviewer 2 Report
The authors overexpressed full-length and soluble Endoglin in different liver cell types and analyzed how different forms are loaded in exosomes. They showed that glycosylated endoglin is efficiently loaded in exosomes.
The weakness of this paper is that it remains unclear the function of endoglin in exosome secreted from different liver cell types.
1. Page 6, lines 256, please start to explain from Supplementary Figure 1.
2. Figure 1C, the expression of SMA appears to be same in lane 3 and 4.
3. Figure 1E, why TGFb does not induce SMA in rat pMF (lane 1 and 2)?
4. Figure 2C, the size of sol-Eng is increased in lane 4 compared to lane 3. Is this difference attributed to the degree of glycosylation?
5. Page 14, lines 508, Supplementary Figure 1 seems to be not relevant to this sentence.